# Poisoning cases and their management in Amhara National Regional State, Ethiopia: Hospital-based prospective study

**Assefa Belay Asrie** [1]*, **Seyfe Asrade Atnafie**[1], **Kefyalew Ayalew Getahun**[1], **Eshetie Melese Birru**[1], **Gashaw Binega Mekonnen**[2], **Geta Asrade Alemayehu**[3], **Berhanu Fikadie Endehabtu**[4], **Marta Berta Badi**[5], **Getinet Mequanint Adinew**[1]

1 Department of Pharmacology, School of Pharmacy, College of Medicine and Health Sciences, University of Gondar, Gondar, Ethiopia, 2 Department of Clinical Pharmacy, School of Pharmacy, College of Medicine and Health Sciences, University of Gondar, Gondar, Ethiopia, 3 Department of Health Service Management and Health Economics, Institute of Public Health, College of Medicine and Health Sciences, University of Gondar, Gondar, Ethiopia, 4 Department of Health Informatics, Institute of Public Health, University of Gondar, Gondar, Ethiopia, 5 Department of Women's and Family Health, School of Midwifery, College of Medicine and Health Sciences, University of Gondar, Gondar, Ethiopia

☯ These authors contributed equally to this work.
* assefabelay@gmail.com

**Data Availability Statement:** All relevant data are within the paper and its Supporting Information file.

## Abstract

### Background

Poisoning is a significant public health problem globally. Ethiopia is a low-income country undergoing technological and social change that may increase access to drugs and chemicals, potentially increasing the incidence of poisoning. This study describes the epidemiology of hospital admissions due to poisoning in a region of Ethiopia.

### Methods

An institution based prospective observational study was employed, as a study design, in selected hospitals of the region from January to December 2018.

### Results

Of 442 poisoning cases, 78 (17.6%) died. Almost all poisoning cases were intentional self-poisonings. The most frequent poisonings were organophosphate compounds, 145 (32.8%), and metal phosphides (majorly aluminum phosphide), 115 (26.0%). The ingested poison was most frequently accessed from the patients' homes, 243 (55.0%), followed by purchases from local shops, 159 (36%). The median duration of admission was 24 hours. Of all the cases, 23 (5.2%) were admitted to intensive care units (ICU) requiring mechanical ventilation. Most of the cases admitted to the ICU were aluminum phosphide-poisoned patients. The majority of deaths (43 of 78) were due to metal phosphides. From the multivariate logistic regression analysis, altered level of consciousness on hospital arrival, metal phosphide poisoning, and no laboratory result as a part of the diagnosis process or

**Funding:** This study was done using a grant awarded by University of Gondar, Ethiopia (Reference № R/T/T/C/Eng.58/03-2018). The funder had no role in the study design, data collection and analysis, decision to publish, or preparation of the manuscript.

**Competing interests:** The authors have declared that no competing interests exist.

investigation of the extent of toxicity were found to be significantly associated with the likelihood of poor treatment outcome.

## Conclusion

The majority of the poisoning cases were females. The most common reasons for the intent of self-poisoning were dispute-related, mainly family disharmonies, followed by psychiatric conditions. The poisoning agents were mostly obtained from households. Organophosphate compounds and metal phosphides were the first and the second most frequently encountered poisoning agents, respectively, and it was noted that the later ones were responsible for most of the fatal cases. Of the pharmacologic interventions, atropine was the only agent regarded as an antidote. The most commonly employed agent for supportive treatment was cimetidine followed by maintenance fluids, while gastric lavage was the only GI decontamination method used among others. The fatality rate of poisoning in this study was found to be much higher than in other similar studies. Impaired consciousness upon hospital arrival, metal phosphide poisoning, and no involvement of laboratory investigation were found to significantly associate with the likelihood of death. Generally, the results dictate the need for the design and implementation of strategies to create awareness, prevent, and manage poisoning incidences in the community.

## Background

Intentional self-poisoning is a common cause of emergency room visits and hospital admissions worldwide and a major reason for illness and death in many countries [1]. Poisoning from pesticides, industrial chemicals, pharmaceuticals, chemical products, and natural toxins is a remarkable public health problem worldwide [2]. An estimate of a systematic review that identified information spanning 108 nations from 2010 to 2014 indicates that there were over 110,000 pesticide-self-poisoning deaths every year, or 13.7% of all suicides worldwide [3]. As advances in technological and social developments have led to ease accessibility of most drugs and chemical substances in the society, the actual number of incidences of poisoning can be higher, because most cases of poisoning actually go unreported [4]. Poisoning was reported as the second most common means of attempted suicide in a district of Ethiopia. The authors also reported that the majority of the poisoning cases were women, and strong detergents and rodenticides were the most frequently used poisons [5]. One prospective study done in selected hospitals in western Ethiopia reported that about 7.1% of poisoning cases died [6]. Similarly, another retrospective study reported that 6.6% of the poisoning cases died [7]. It can be depicted from the reports that poisoning is one of the health problems contributing to the morbidity and mortality burdens in the country.

The types and frequencies of poisoning agents encountered vary considerably across the world and even in different parts of the same country due to socioeconomic and cultural differences. The frequency of poisoning incidences in a given community depends on the availability of poisonous agents, the common occupational activity in that society, and religious and sociocultural influences [8–10]. Household chemicals and prescribed drugs are the most common poisoning agents in the developed world, while agricultural chemicals are the most common poisoning agents in developing countries [8]. More importantly, limited drug and chemical regulatory activities, inadequacy of surveillance systems, poor enforcement, and ease

of access to various kinds of drugs or chemicals have been implicated in the higher poisoning prevalence and case fatality in derdeveloping countries [11, 12].

This study was carried out to determine the frequently encountered poisoning agents, the clinical management, and the patient outcome (as cure or death) and associated factors (S1 File).

## Methods

### The study area and period

The Amhara National Regional State is one of the 10 regional states in Ethiopia. There were more than 70 hospitals, 100 health centers, and 700 health stations in the region during the proposal development for this study. Five of the hospitals were referral-level hospitals. Four of the referral hospitals were conveniently selected as the study sites. The hospitals included in the study are University of Gondar Comprehensive Specialized Hospital, Felege Hiwot Referral Hospital, Debre Markos Referral Hospital, and Dessie Referral Hospital. University of Gondar Comprehensive Specialized Hospital is located in Gondar Town, 750 km northwest of the capital city of Ethiopia, Addis Ababa. It serves a catchment population of approximately 5 million people in Central Gondar and the neighboring zones. Felege Hiwot Referral Hospital is found in Bahir Dar City, which is 562 km northwest of Addis Ababa. The hospital serves a catchment population of more than 5 million. Debre Markos Referral Hospital is located in Debre Markos Town, 300 km northwest of Addis Ababa. The hospital serves more than 3.5 million people in East Gojjam Zone and neighboring zones. Dessie Referral Hospital is located in Dessie Town, in the northern of Ethiopia, at a distance of 400 km from Addis Ababa. It provides healthcare services for about 9 million people in the South Wollo Zone of the Amhara region and the surrounding areas. At all sites, the data collection was started on January 1, 2018 and completed in the last week of December 2018 on different specific dates. Nearly a year was needed to recruit the designated number of cases at each site.

### Study design and sampling technique

An institution-based prospective observational study was employed, as a study design, at the selected hospitals in the region. Considering the number of referral hospitals and their distribution in the region, four referral hospitals were conveniently selected for the study. All consecutive cases admitted to the emergency departments of the hospitals during the data collection period were included in the study, regardless of their age and causes of poisoning. The cases that were not willing to participate in the study and from whom adequate data were not obtained were excluded from the study.

### Sample size determination

The sample size was determined by using a single population proportion formula considering the following assumptions: prevalence (p) of poisoning cases 50% to get the larger sample size, standard normal distribution value at 95% confidence level of Z/2 = 1.96, and margin of error (w) = 5%. This gives a sample size of 384 patients, as shown below.

$$N = \frac{Z^2 p(1-p)}{w^2} = \frac{(1.96)^2 0.5(1-0.5)}{(0.05)^2} = 384.16 \approx 384 \text{ patients}$$

Where, N is the sample size, Z is confidence interval, p is a proportion, and w is the margin of error.

Once the minimum sample size was obtained in this way, by adding a 15% non-response rate, the total sample size was determined to be 442. The final sample was then distributed to

the four study sites: 110 cases to each of University of Gondar Comprehensive Specialized Hospital and Felege Hiwot Referral Hospital and 111 cases to each of Debre Markos Referral Hospital and Dessie Referral Hospital. Lottery method was used to decide how many cases would be recruited at each site. Since the data was collected until we achieved the predetermined number of subjects in each study site, the total number of cases targeted is equal to the originally calculated sample size.

## The data collection tool, process, and quality control

Data was collected using a structured questionnaire. The questionnaire was designed to collect data regarding the patients' socio-demographic details (sex, age, ethnicity, occupation, educational status, and marital status) and facts related to the poisoning, such as the type of poison used, reasons for deliberate self-poisoning, the clinical state of the patient on presentation at the emergency department, and the treatment and outcome of the case. At each site, the data were collected by two trained nurses working in the emergency department of the hospital. The data collection process involved filling out the questionnaire during diagnosis and reviewing the patient record chart. To ensure the quality of the data collected, the data collectors were trained on the overall study design and objectives, the contents of the questionnaire, and approaching interviewees. Moreover, supervisors were assigned from the study team for periodic monitoring of the data collection process. During supervision, some filled-out questionnaires were randomly selected and cross-checked with their respective patient medical records. Furthermore, each filled-out questionnaire was checked for its completeness and clarity before received from the data collectors.

## Data entry, analysis, and interpretation

The data was entered into SPSS Version 20 and analyzed using statistical tests. The interpretations were done using descriptive statistics such as percentages and frequency distributions. Furthermore, the impacts of potential predictors of treatment outcomes (cured or died) were estimated using binary logistic regression analysis at a cut-off point of $p < 0.05$ for statistically significant association and a confidence interval (CI) of 95%. In this regard, multivariate analysis was carried out for predictor variables using backward stepwise multiple logistic regression. The adjusted odds ratio (AOR) and its 95% CI were calculated for each predictor variable included in the ananalysis. The variables found to be significantly associated with the likelihood of death were considered the risk factors for poor treatment outcomes from poisoning.

## Ethics approval and consent to participants

The protocol of the study was reviewed and approved by the Institutional Ethical Review Board of University of Gondar (Reference №: O/V/P/RCS/05/372/2018). In addition, each hospital administration office was requested by a formal letter from the University of Gondar, College of Medicine and Health Sciences, Research and Community Service Office, and permission was obtained from each hospital to conduct the study. Furthermore, written informed consent was obtained from each patient or caregiver for inclusion in the study. Those who were unwilling were excluded. Furthermore, no personal identifier was included in the data collection tool, and the data record was kept confidential and used only for the study purposes.

## Results

### Cases excluded

A total of 463 patients were approached during the data collection period. However, 21 of the patients were subsequently excluded from participation. Thirteen of them were excluded for

the incompleteness of the data they provided, while the remaining eight declined to provide consent. Nevertheless, as data collection continued until the predetermined number of subjects was reached at each study site, the total number of cases recruited and ultimately included in the study remained consistent with the originally calculated sample size.

## Socio-demographic characteristics of the participants

The majority of the patients were under 40 years of age, females, Orthodox Christians, and Amhara in ethnicity. About half of the cases were single in their marital status. A slightly higher percentage of the cases were from rural residential areas. A relatively higher proportion of the patients could not read and write, followed by those at the secondary school level. Occupationally, students and framers were the first- and second-most frequent cases (Table 1).

## Types of poisoning agents and death distributions

Organophosphate compounds and metal phosphides were the first and the second most commonly encountered causes of poisoning, respectively. The total fatality rate was found to be 17.6%. The majority of fatal cases were among metal phosphide (43 of 78) and organophosphate compound (25 of 78) poisoned cases (Table 2).

**Table 1. Socio-demographic characteristics of the participants.**

| Variables | | Frequency | % |
|---|---|---|---|
| Sex | Male | 163 | 36.9 |
| | Female | 279 | 63.1 |
| Age | ≤20 years | 147 | 33.3 |
| | 21–40 years | 235 | 53.2 |
| | >40 years | 60 | 13.6 |
| Religion | Orthodox Christian | 363 | 82.1 |
| | Muslim | 77 | 17.4 |
| | Protestant | 2 | 0.5 |
| Ethnicity | Amhara | 434 | 98.2 |
| | Others | 8 | 1.8 |
| Residence | Rural | 216 | 48.9 |
| | Urban | 226 | 51.1 |
| Marital status | Married | 201 | 45.5 |
| | Single | 223 | 50.5 |
| | Other | 18 | 4.1 |
| Educational status | Cannot read and write | 128 | 29.0 |
| | Can read or write | 72 | 16.3 |
| | Primary school | 67 | 15.2 |
| | Secondary school | 116 | 26.2 |
| | College level | 43 | 9.7 |
| | University level | 16 | 3.6 |
| Occupation | Student | 139 | 31.4 |
| | Farmer | 130 | 29.4 |
| | Self-employee | 91 | 20.6 |
| | House wife | 26 | 5.9 |
| | Government employee | 20 | 4.5 |
| | Daily laborer | 18 | 4.1 |
| | Others | 18 | 4.1 |

**Table 2. Poisoning agents encountered and patient outcomes.**

| Poisoning agent | Treatment outcomes, n (%)* | | Total, n (%)* |
|---|---|---|---|
| | **Survived** | **Died** | |
| Organophosphates | 120 (27.1) | 25 (5.7) | 145 (32.8) |
| Metal phosphides | 72 (16.3) | 43 (9.7) | 115 (26.0) |
| Bleaching agents | 44 (10.0) | 0 (0.0) | 44 (10.0) |
| Medicines | 31 (7.0) | 2 (0.5) | 33 (7.5) |
| Rat poison | 22 (5.0) | 4 (0.9) | 26 (5.9) |
| Alcohol | 20 (4.5) | 0 (0.0) | 20 (4.5) |
| Carbon monoxide | 14 (3.2) | 0 (0.0) | 14 (3.2) |
| Food poisoning | 12 (2.7) | 0 (0.0) | 12 (2.7) |
| Unknown chemicals | 6 (1.4) | 1 (0.2) | 7 (1.6) |
| Herbs | 5 (1.1) | 0 (0.0) | 5 (1.1) |
| Snake bite | 5 (1.1) | 0 (0.0) | 5 (1.1) |
| Dichlorodiphenyltrichloroethane (DDT) | 4 (0.9) | 0 (0.0) | 4 (0.9) |
| Others# | 9 (2.0) | 3 (0.7) | 12 (2.7) |
| Total, n (%)* | 364 (82.4) | 78 (17.6) | 442 (100.0) |

#Include fertilizer, unspecified herbicides & insecticides, and match head

*The percentages are calculated out of all cases.

## Sources of the poisoning agents

Home was reported by most of the poisoned patients as a source of poisoning agents (55.0%), followed by shops (36.0%). The other sources reported include pharmacies, veterinary drug venders, workplaces, fire accident smoke, traditional medical practitioners (Fig 1).

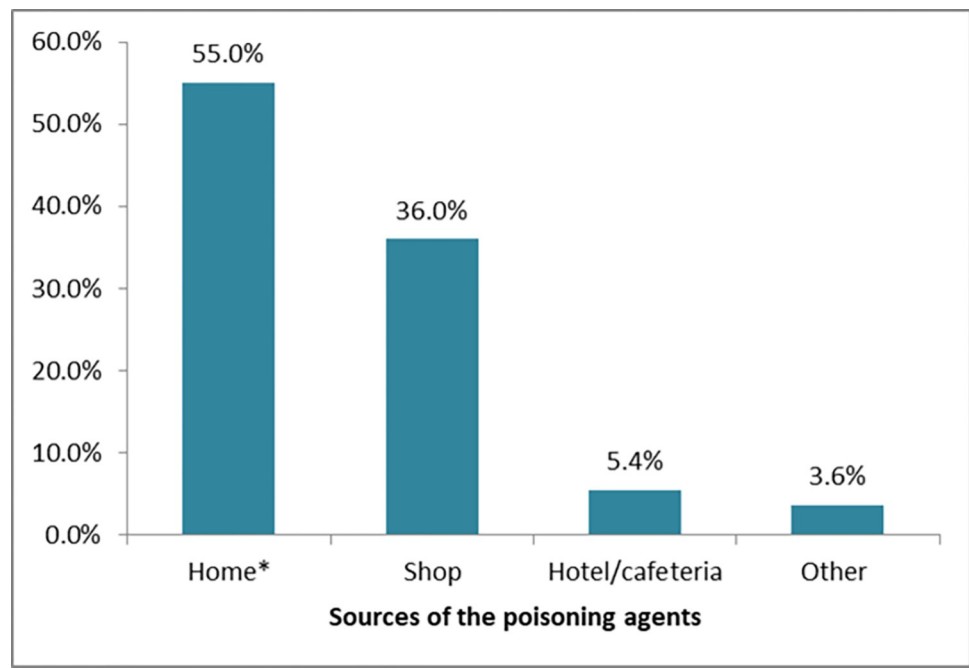

**Fig 1. The percentage distribution of the source of poisoning agents encountered.** *Includes homes other than the cases were living in.

**Table 3. Reported reasons behind the poisoning incidents.**

| Reason | Frequency | % |
|---|---|---|
| Dispute related[#] | 205 | 46.4 |
| Psychiatric disorders | 58 | 13.1 |
| Love affairs | 45 | 10.2 |
| Unknown | 34 | 7.7 |
| Accident | 32 | 7.2 |
| Joblessness | 18 | 4.1 |
| Disease treatment | 11 | 2.5 |
| Recreational purpose | 11 | 2.5 |
| Unplanned pregnancy | 8 | 1.8 |
| Loss of beloved ones | 7 | 1.6 |
| Academic failure | 6 | 1.4 |
| Others | 7 | 1.6 |

[#]Most dispute related reasons were disputes within families.

## Reasons for poisoning incidents

Family disputes (46.4%), psychiatric illnesses (13.1%), and love affairs (10.2%) were the top three frequently reported reasons for self-poisoning. In addition to the specified reasons mentioned, 7.7% of the patients were poisoned for unknown reasons (Table 3).

## Treatments

The sole proven antidote among the treatments mentioned was atropine, which was utilized in 22.4% of the instances. Magnesium sulfate was indicated in 17.6% of the cases. Gastric lavage was the only gastrointestinal decontamination method used in managing 17.0% of the cases. Maintenance fluids and cimetidine were the first and second most commonly used agents for supportive or symptomatic treatment of the poisoning cases. Additionally, of the total cases in this study, 5.2% were admitted to intensive care units (ICU) and intubated or mechanically ventilated, or given circulatory support. Most of the cases admitted to the ICU were aluminum phosphide-poisoned patients (data not shown). The patients were admitted to the ICU because of serious symptoms, particularly those of respiratory failure and severe hypotension that, necessitated endotracheal intubation or mechanical ventilation and/or circulatory support (Table 4).

## Factors associated with treatment outcome

A bivariate analysis was done for each potential variable assumed to be associated with the treatment outcome. The variables that were verified to be significantly associated with treatment outcomes in the bivariate analysis were included in the multivariate logistic regression model. Based on the thier level of consciousness upon arrival at the emergency departments of the hospitals, those who were in a semiconscious state were 12.145 times more likely to die (AOR: 12.145, 95% CI: 3.686–40.019, $p < 0.001$) than those who were in a conscious state. Moreover, those who were in an unconscious state at arrival were 38.423 times more likely to die (AOR: 38.423, 95% CI: 10.408–141.848, $p < 0.001$) than those were in a conscious state. An hour increase in the time of hospital stay decreased the likelihood of death by 9.1% ($\beta = -0.096$, AOR: 0.909, 95% CI: 0.882–0.936, $p < 0.001$). Patients poisoned with metal phosphides were 6.983 times (AOR: 6.983, 95% CI: 2.559–19.056, $p < 0.001$) more likely to die than those

Table 4. Treatments given to the poisoned patients[♯].

| Agents/method | | Frequency | % |
|---|---|---|---|
| **Antidotal** | Atropine | 99 | 22.4 |
| | Magnesium sulfate* | 78 | 17.6 |
| **GI decontamination** | Gastric lavage | 75 | 17.0 |
| **Removal enhancer** | Frusemide | 28 | 6.3 |
| **Supportive/symptomatic** | Maintenance fluids | 239 | 54.1 |
| | Cimetidine | 267 | 60.4 |
| | Antibiotics | 54 | 12.2 |
| | Hydrocortisone | 46 | 10.4 |
| | Omeprazole | 41 | 9.3 |
| | Metoclopramide | 25 | 5.7 |
| | Dopamine | 21 | 4.8 |
| | Vitamin B complex | 16 | 3.6 |
| | 40% glucose | 15 | 3.4 |
| | Calcium gluconate | 10 | 2.3 |
| | Oral rehydration salts (ORS) | 8 | 1.8 |
| | Tramadol | 6 | 1.4 |
| | Antacids | 5 | 1.1 |
| **Admitted to ICU** | Mechanical ventilation and other supportive cares | 23 | 5.2 |

[♯]Agents with indication frequency of less than five have been excluded.

*Mechanistically has antidotal activity but not approved yet.

intoxicated with organophosphate agents. In addition, the odds of death among poisoned patients who had no laboratory result as a part of their diagnosis process or investigation of extent of toxicity was 5.376 times (AOR: 5.376, 95% Cl: 1.633–17.704, $p < 0.01$) than those whose diagnosis involved laboratory tests (Table 5).

## Discussion

This study investigated poisoning patterns in relation to socio-demographic characteristics, the type of poisoning agents involved, treatments given and associated factors with patient outcomes among poisoning cases attended the selected referral hospitals in Amhara National Regional State, Ethiopia. The main findings of the study are explained and/or compared with previously published findings of similar studies.

The incidence of poisoning may vary with different categories of socio-demographic characteristics such as age group, sex, residence, and occupation. In this study, the majority of the poisoning cases were females (63.1%). This result is consistent with the results of previous studies in Ethiopia and Taiwan, which reported 59.20% and 54.76% of incidences, respectively, were females [6, 13]. More than half (53.2%) of the cases were in the age group of 21–40 years, and this is in line with the results of other similar studies [7, 14]. Regarding residence, the poisoning incidences were found to be somewhat greater in urban areas (51.1%). This difference might be, at least in part, because some of the cases from rural areas may not attend hospitals as they may be treated in health posts, health centers, or district hospitals in their locality. Considering the distribution of the cases by marital status, 50.5% of the poisoned patients were single. This proportion is lower than reports of similar studies in Addis Ababa, Ethiopia and Selected Hospitals in Western Ethiopia, which were 54.98% and 57.10%, respectively [6, 15]. The largest proportion of the poisoning cases were students (31.4%), followed by farmers

**Table 5. Multivariate analysis of factors associated with treatment outcome ($R^2$ = 0.654, Hosmer and Lemeshow goodness of Fit Test, p = 0.924).**

| Explanatory Variables | β-coefficient | Treatment outcome | | AOR (95% C.I.) | p-value |
|---|---|---|---|---|---|
| | | Cured | Died | | |
| **Age (year)** | 0.032 | - | - | 1.033 (0.997–1.069) | 0.071 |
| **Residence** | | | | | 0.215 |
| Rural | - | 169 | 47 | 1 | |
| Urban | 0.693 | 195 | 31 | 0.657 (0.338–1.275) | |
| **Level of consciousness[1]** | | | | | 0.000 |
| Conscious | - | 189 | 19 | 1 | |
| Semiconscious | 2.497 | 88 | 19 | **12.145 (3.686–40.019)*** | |
| Unconscious | 3.649 | 87 | 40 | **38.423 (10.408–141.848)*** | |
| **Distance (km)[2]** | 0.004 | - | - | 1.004 (.986–1.022) | 0.686 |
| **Time of arrival (hr)[3]** | -0.040 | - | - | 0.961 (0.890–1.038) | 0.312 |
| **Duration of hospital stay** | -0.096 | - | - | **0.909 (0.882–0.936)*** | 0.000 |
| **Major poison groups** | | | | | 0.000 |
| Organophosphates | - | 120 | 25 | 1 | |
| Metal phosphides | 1.944 | 72 | 43 | **6.983 (2.559–19.056)*** | |
| **Laboratory test** | | | | | 0.006 |
| Yes | - | 111 | 11 | 1 | |
| No | **1.682** | 253 | 67 | **5.376 (1.633–17.704)** | |

The cut point for significant association is at p < 0.05.

*p < 0.05

**p < 0.01

***p < 0.001

[1]Level of consciousness deermined just on arrival at the referral hospitals

[2]The distance is from the place of exposure at the referral hospital.

[3]Time of arrival is the time from the time of exposure to arrival to the referral hospitals.

(29.4%) in occupation. This may be because students were more prone to emotional liability associated with academic failure and love-related issues, leading to intentional poisoning for suicidal purposes.

Organophosphate compounds and metal phosphides (mainly aluminum phosphide) were the first and second most commonly reported causes of poisoning, followed by other agents such as bleaching agents, medicines, unknown chemicals regarded as rat poison, alcohol, and carbon monoxide in decreasing order of frequency. Concerning organophosphates, this result is consistent with the findings of other studies reporting these compounds as the most frequently encountered poisoning agents [14, 16–18], but inconsistent with another study at Jimma University Specialized Hospital [10]. In Ethiopia, organophosphate compounds and aluminum phosphide are widely available as pesticides. This may increase their availability and ease their accessibility for intentional poisoning.

Concerning the sources of the poisoning agents, the majority were obtained at residential homes and local shops. This result is in agreement with the results of other studies in that most of the poisoning agents were obtained at home [7, 15]. This could be because of the inappropriate storage of the poisoning agents at home when leftover from their use as insecticides or against grain storage pastes. Shops were found to be the second most frequently reported source of poisoning agents. This finding aligns with the result of a similar study conducted in western Ethiopia [9]. The most common reasons for intentional poisoning were dispute-related, mainly family disharmonies, followed by psychiatric conditions, love affairs, and

unidentified causes, among others. The reasons reported in this study are similar to those in a systematic review by Chelkeba et al [19]. Furthermore, regarding the most frequently reported reason for poisoning, the result of this study is in line with other previous studies [16, 20].

Various interventions were used to treat the poisoning cases. The sole proven antidote among the pharmacologic treatments mentioned was atropine, known to be an antidote to organophosphate poisoning. Magnesium sulfate has been reported to have beneficial effects on aluminum phosphide and organophosphate poisonings [21, 22]. Furthermore, preclinical and clinical findings indicate that magnesium sulfate could be a promising supplementary treatment for acute organophosphorus insecticide poisoning. Nevertheless, the available evidence is presently insufficient to warrant its recommended utilization [23].The remaining pharmacologic agents are used more for supportive treatment of poison-related symptoms than for antidotal effects. The most commonly employed nonpharmacologic interventions were supportive care (ABC, mainly IV resuscitation), and this finding is favorable as ensuring a protected airway, adequate ventilation, and hemodynamic stability through the ABC approach is crucial in the management of poisoning emergencies [19].

In this study, only gastric lavage was found to be employed for GI decontamination among the possible GI decontamination methods. Gastric lavage is indicated for immediate removal of stomach contents after an orally ingested drug overdose or poisoning [24]. The use of this method in about 17.0% of the cases in this study was most probably from this perspective. However, it is not recommended for routine use and should not be considered unless a patient has ingested a potentially life-threatening amount of poison [25]. It should be considered only when there is a risk of significant toxicity or imminent fatality and in situations where antidotal or other supportive modalities are inadequate after contraindications are ruled out [24]. Because its efficiency in removing the poisoning agents was found to be highly variable and diminished with time in experimental and clinical studies, it showed limited beneficial effects. Moreover, the procedure is associated with serious risks, including hypoxia, dysrhythmias, perforation of the GI tract or pharynx, aspiration pneumonitis, and others [26]. Therefore, consideration should first be given to other less invasive modalities of gastrointestinal decontamination, such as activated charcoal or whole-bowel irrigation [24]. However, there was no report of using the other GI decontamination agents or methods. Among all the cases, dialysis was used only in one case. In essence, this suggests that it was more likely due to the unavailability of functional dialysis facilities or the unaffordability of the procedure than that poisoning cases needing dialysis for poison removal were rare.

The case fatality rate from acute poisoning was found to be 17.6%. This finding is much higher than previous findings of similar studies carried out in Ethiopia [6, 9, 13, 15, 27] and India [14], ranging from 4.10 to 10.20%. Out of 78 deaths, metal phosphides were responsible for 55.1% of the deaths, whereas organophosphates were attributed to 32.1% of the deaths. Even though bleaching agents were the third most common cause of poisoning, no death was recorded among the victims poisoned by these agents. Organophosphates, aluminum phosphide, and other chemicals are available in Ethiopia for use as rodenticides, insecticides, or herbicides [28]. However, a comparative study showed that Ethiopian officers who handle these chemicals are less knowledgeable about the products [29].

Aluminum phosphide is widely available in Ethiopia, with approved use for the control of maize weevils and flour beetles on stored maize [28]. Organophosphate compounds, especially malathion, are also among the commonly available pesticides for agricultural uses [29]. Because of their wide availability and ease of accessibility, plus their inherent toxicity, these agents are more likely to pose a great health burden to the community. This result underscores the necessity of developing and implementing strategies aimed at raising awareness and

preventing poisonings resulting from either intentional consumption or accidental exposure of the commonly encountered poisoning agents within the community.

Multivariate logistic regression analysis was performed to identify predictors of treatment outcome in the present study. In the analysis, patient level of consciousness, length of hospitalization, type of poisoning agents encountered, and use of laboratory tests in patient diagnoses were found to be significant predictors of treatment outcome. The patients who were in semi-conscious and unconscious states upon hospital arrival were 12.145 and 38.423 times more likely to die, respectively, than those who were conscious. The level of consciousness may be compromised due to a delay in intervention or because of the inherent toxic characteristics of a poisoning agent, leading to rapid loss of consciousness and negatively affecting the treatment outcome. The analysis showed that as the length of hospitalization increased, the probability of death decreased. This could be because the likelihood of death decreased once the patients managed to overcome acute symptoms and received care after being admitted to the hospitals. There was no discernible relationship between distance and treatment outcome. It makes sense that the treatment outcome would be poor consistently the further a patient traveled to get to the hospital. The findings of the present study in this regard could be the consequence of pre-hospital interventions carried out at home, community health centers, or smaller general hospitals, which were not taken into account. Patients who were poisoned with metal phosphides exhibited a mortality risk 6.983 times higher ($p < 0.001$) compared to individuals intoxicated with organophosphate agents. Moreover, 43 of the 78 fatal cases were among metal phosphide-poisoned patients. This suggests that metal phosphide poisoning may present a graver prognosis, indicative of its heightened severity, thus warranting significant concern as a serious public health threat within the community. The liklihood of death among poisoned patients whoes diagnosis did not involve laboratory testing was 5.376 times higher compared to those who did receive such tests in their diagnosis.This might be because the laboratory diagnosis can help investigate the severity of toxicity and direct treatment intervention accordingly.

The prospective nature of the study design enables us to collect detailed clinical and laboratory data to determine the severity of the cases as a determinant of treatment outcome. This can be taken as the strength of this study. However, the study lacks some parameters including pre-hospital interventions (that might be given at home or local health facilities), referral cases, and appropriateness of interventions. The absence of these variables can be considered a limitation of this study. Especially, the result of this study may be affected by referral bias. If these variables were included, it would be better for a more appropriate explanation of potential predictor variables of patient outcome. For example, the likelihood of death was not significantly increased with the time of delay for hospital arrival, and this might be because of pre-hospital interventions at home or local health centers. This does not imply that delays in reaching hospitals after intoxication have no negative impact on treatment outcomes. Similarly, the inconsistent association between the distances from the areas where the poisoning incidence occurred to the hospitals would be more objectively explained if the possible pre-hospital interventions and referral cases were considered.

## Conclusion

The majority of the poisoning cases were females. The most common reasons for the intent of self-poisoning were dispute-related, mainly family disharmonies, followed by psychiatric conditions. The poisoning agents were mostly obtained from households. Organophosphate compounds and metal phosphides were the first and the second most frequently encountered poisoning agents, respectively, and it was noted that the later ones were responsible for most

of the fatal cases. Of the pharmacologic interventions, atropine was the only agent regarded as antidote. The most commonly employed agent for supportive treatment was cimetidine followed by maintenance fluids, while gastric lavage was the only GI decontamination method used among others. The fatality rate of poisoning in this study was found to be much higher than in other similar studies. Impaired consciousness upon hospital arrival, metal phosphide poisoning, and no involvement of laboratory investigation were found to significantly associate with the likelihood of death. Generally the results dictate the need for the design and implementation of strategies to create awareness, prevent, and manage poisoning incidences in the community.

## Supporting information

**S1 File. The supporting information includes S1-S6 Tables.** The distribution of patient outcomes in the categorical variables, which are not included in the multivariate analysis model. Besides, it provides the details of the patien outcomes with the continuous variables that appeared in the multivariate analysis model.
(DOCX)

## Acknowledgments

We are grateful to the administration offices of the hospitals for their cooperation in facilitating the official process of getting permission to undertake the study at their respective institutions. Besides, we are thankful to the physicians and nurses who were working in the emergency departments of the hospitals for their unwavering cooperation in the data collection process.

## Author Contributions

**Conceptualization:** Assefa Belay Asrie, Kefyalew Ayalew Getahun, Eshetie Melese Birru, Gashaw Binega Mekonnen, Geta Asrade Alemayehu, Berhanu Fikadie Endehabtu, Marta Berta Badi, Getinet Mequanint Adinew.

**Data curation:** Assefa Belay Asrie.

**Project administration:** Assefa Belay Asrie, Getinet Mequanint Adinew.

**Supervision:** Assefa Belay Asrie, Seyfe Asrade Atnafie, Kefyalew Ayalew Getahun, Eshetie Melese Birru, Gashaw Binega Mekonnen, Geta Asrade Alemayehu, Berhanu Fikadie Endehabtu, Marta Berta Badi, Getinet Mequanint Adinew.

**Writing – original draft:** Assefa Belay Asrie.

**Writing – review & editing:** Seyfe Asrade Atnafie.

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
