## [Decision Letter · Decision Letter 0]

31 Oct 2022

PONE-D-22-26894Poisoning Cases and Their Management in Amhara Region, Ethiopia: Hospital Based Prospective StudyPLOS ONE

Dear Dr. Asrie,

Thank you for submitting your manuscript to PLOS ONE. After careful consideration, we feel that it has merit but does not fully meet PLOS ONE’s publication criteria as it currently stands. Therefore, we invite you to submit a revised version of the manuscript that addresses the points raised during the review process.

We look forward to receiving your revised manuscript.

Kind regards,

Senthil Kumaran, MBBS, MD, DNB

Academic Editor

PLOS ONE

Journal Requirements:

"This study was done using a grant awarded by University of Gondar, Ethiopia, Reference № R/T/T/C/Eng.58/03-2018"

Please state what role the funders took in the study.  If the funders had no role, please state: ""The funders had no role in study design, data collection and analysis, decision to publish, or preparation of the manuscript."" If this statement is not correct you must amend it as needed. 

"We would like to thank University of Gondar for its financial support which was used for the data collection, supervision, and related costs of this study."

"This study was done using a grant awarded by University of Gondar, Ethiopia, Reference № R/T/T/C/Eng.58/03-2018"

"We have no competing interests regarding this work"

7. Please amend the manuscript submission data (via Edit Submission) to include author "Seyfe Asrade Atnafie". 

**Additional Editor Comments:**

The paper can be accepted if only following queries by the reviewers is done accordingly.

Reviewer 1:

1. This manuscript requires spelling corrections and grammatical corrections.

2. The spelling mistakes and the few of the required grammatical corrections are commented and highlighted in the attached manuscript itself.

Reviewer 2:

1. Percentage would be rounded off to the first decimal place.

2. What is “DDT”? “DDT” should be spelled fully on the first mention.

3. How many patients were intubated or ventilated mechanically? The description about intubation and mechanical intubation would be helpful.

4. What was the indication of gastric lavage in this study?

5. What was the indication of frusemide and vitamin B complex?

6. Were there admission cases to the hospital without IV fluid in this study?

7. What kind of laboratory tests in poisoning test do you have in this study? CBC, ECG, or blood toxicant concentration? What was the indication of laboratory test in poisoning cases?

8. Authors stated that the distance from the place of poisoning to hospital (10-20 and >40km) was associated with treatment outcome. Why was not associated with 20-30 or 30-40km? The discussion about that differences would be helpful.

Reviewers' comments:

Reviewer's Responses to Questions

**Comments to the Author**

1. Is the manuscript technically sound, and do the data support the conclusions?

Reviewer #1: Yes

Reviewer #2: Yes

2. Has the statistical analysis been performed appropriately and rigorously? 

Reviewer #1: N/A

Reviewer #2: Yes

3. Have the authors made all data underlying the findings in their manuscript fully available?

Reviewer #1: Yes

Reviewer #2: Yes

4. Is the manuscript presented in an intelligible fashion and written in standard English?

Reviewer #1: No

Reviewer #2: No

5. Review Comments to the Author

Reviewer #1: This manuscript requires spelling corrections and grammatical corrections. The spelling mistakes and the few of the required grammatical corrections are commented and highlighted in the attached manuscript itself. Authors may relook at entire manuscript for spelling mistakes and grammatical corrections.

Reviewer #2: Thank you for giving the opportunities to review this manuscript.

Authors reviewed the 442 poisoning cases for two years in Ethiopia. The majority of causes were organophosphate and metal phosphides. Older ages, impaired consciousness on arrival, distance from the institution, and no laboratory test were found to be associated with the mortality or poor outcome. This study had important observations, but please addresses all my concerns below for a potentially better paper:

1. Percentage would be rounded off to the first decimal place.

2. Causes of poisoning (P9)

What is “DDT”? “DDT” should be spelled fully on the first mention.

3. How many patients were intubated or ventilated mechanically? The description about intubation and mechanical intubation would be helpful.

4. What was the indication of gastric lavage in this study?

5. What was the indication of frusemide and vitamin B complex?

6. Were there admission cases to the hospital without IV fluid in this study?

7. What kind of laboratory tests in poisoning test do you have in this study? CBC, ECG, or blood toxicant concentration? What was the indication of laboratory test in poisoning cases?

8. Authors stated that the distance from the place of poisoning to hospital (10-20 and >40km) was associated with treatment outcome. Why was not associated with 20-30 or 30-40km? The discussion about that differences would be helpful.

6. PLOS authors have the option to publish the peer review history of their article (what does this mean?). If published, this will include your full peer review and any attached files.

Reviewer #1: No

Reviewer #2: No

---

## [Author Response · Author response to Decision Letter 0]

18 Nov 2022

Response to Editor 

Manuscript Title: Poisoning Cases and Their Management in Amhara Region, Ethiopia: Hospital Based Prospective Study

Manuscript Submission Number: PONE-D-22-26894

Journal: PLOS ONE

Response to Editor 

Dear Editor, 

We are very thankful to you for handling the publication process of this manuscript and your unreserved and indispensable inputs which help improve it and keep the high standards of the journal. The changes made or the new texts included in the revision process based on your and the reviewers’ comments are marked-up with greed color. Deletions are also shown by red colored texts with Strikethrough line, example. 

Dear Dr. Asrie,

Thank you for submitting your manuscript to PLOS ONE. After careful consideration, we feel that it has merit but does not fully meet PLOS ONE’s publication criteria as it currently stands. Therefore, we invite you to submit a revised version of the manuscript that addresses the points raised during the review process.

Not applicable 

Journal Requirements:

It meets PLOS ONE’s style requirements. 

"This study was done using a grant awarded by University of Gondar, Ethiopia, Reference № R/T/T/C/Eng.58/03-2018"

Please state what role the funders took in the study. If the funders had no role, please state: ""The funders had no role in study design, data collection and analysis, decision to publish, or preparation of the manuscript."" If this statement is not correct you must amend it as needed. 

The financial disclosure is restated in the revised manuscript as “This study was done using a grant awarded by University of Gondar, Ethiopia (Reference № R/T/T/C/Eng.58/03-2018). The funder had no role in the study design, data collection and analysis, decision to publish, or preparation of the manuscript.”

"We would like to thank University of Gondar for its financial support which was used for the data collection, supervision, and related costs of this study."

"This study was done using a grant awarded by University of Gondar, Ethiopia, Reference № R/T/T/C/Eng.58/03-2018"

The funder had been removed from the acknowledgement section and the 

acknowledgement has been restated as follows. 

“We are grateful to the administration offices of the hospitals for their cooperation in facilitating the official process of getting permission to undertake the study at their respective institutions. Besides, we are thankful to the physicians and nurses who were working in the emergency departments of the hospitals for their unwavering cooperation in the data collection process.” 

"We have no competing interests regarding this work"

In the revised manuscript, the competing interest restated as “The authors have declared that no competing interests exist.” 

We will update your Data Availability statement on your behalf to reflect the information you provide. Thank you! 

In line to the second prompt, there is no legal or ethical restriction on sharing the data set of this study publicly and is submitted as Supporting Information files in the submission of the revised manuscript.

The data set is provided with the revised manuscript, labeled as supporting information. We have no repository information to provide. 

7. Please amend the manuscript submission data (via Edit Submission) to include author "Seyfe Asrade Atnafie". 

It has been amended.

We couldn’t access references 2 & 5. Reference 2 replace by 1new reference and reference 5 by 2 new references. Because of this change, references 6-28 →7-29. 

Additional references are included in the revised manuscript. We have added some explanations to the discussion part based on the comments forwarded by “Reviewer 2”. Some references were used in the process and are cited in the revised manuscript. So there are reshufflings made in the order of the references based on 8 references added and the references used to replace references 2 & 5. The changes are clearly indicated in the revised manuscript reference lists. 

References 30, 31, 32, 33, 34, 35, 36, and 38 are newly included references. In doing so the, reference 29 in the original submission shifted to 37, in the revised manuscript, and references 30-34 to 39-43. 

Additional Notes 

− The values in figure 1 are expressed in percentage in the revised manuscript as we feel that it would be more appropriate. 

− We have made additional corrections/modifications (in addition to the changes based on the concerns of the reviewers or the editor). All changes are shown in the revised manuscript with highlights. 

Additional Editor Comments:

The paper can be accepted if only following queries by the reviewers is done accordingly.

We have addressed the queries by reviewers accordingly. 

Response to Reviewers

Response to Reviewer 1

Thank you so much, dear reviewer, for your valuable comments. 

1. This manuscript requires spelling corrections and grammatical corrections.

We have gone through the manuscript document and made corrections accordingly. Corrections are shown by green color highlight in the revised manuscript.

2. The spelling mistakes and the few of the required grammatical corrections are commented and highlighted in the attached manuscript itself.

Thank you. 

Response to Reviewer 2

Dear Reviewer, 

Thank you dear review for your time and effort invested to review this manuscript. The comments and questions raised are very important to improve the manuscript. The comments and/or the questions are addressed point-by-point as follows. 

1. Percentage would be rounded off to the first decimal place. 

All percentage values are rounded off to the first decimal place as you suggested.

2. What is “DDT”? “DDT” should be spelled fully on the first mention.

Right, it is the abbreviation for “dichlorodiphenyltrichloroethane”. In the revised manuscript, we mentioned in the form of “dichlorodiphenyltrichloroethane (DDT)” as it first appeared and “DDT” for any that comes next. 

3. How many patients were intubated or ventilated mechanically? The description about intubation and mechanical intubation would be helpful.

Orotracheal intubation is the technique of choice to maintain airway patency during respiratory assistance (1). Furthermore, invasive mechanical ventilation is considered the main stay of respiratory life support for severely ill victims (2). Of the total cases in this study, 5.0% were admitted to intensive care unit (ICU) and intubated or mechanically ventilated and/or given circulatory support. The patients were admitted to the intensive care unit (ICU) because of serious symptoms, particularly those of respiratory failure and severe hypotension necessitating endotracheal intubation or mechanical ventilation and/or circulatory support.

Even though the data set at hand does not show the exact number of patients received each intervention, it can be concluded that those patients admitted in the ICU were intubated or mechanically ventilated and/or got cardiovascular monitoring and support.

1. Mégarbane B, Donetti L, Blanc T, Chéron G, Jacobs F. ICU management of severe poisoning with medications or illicit substances. Réanimation. 2006;15(5):343-353.

2. Fialkow L, et al. Mechanical ventilation in patients in the Intensive care unit of a general university hospital in southern Brazil: an epidemiological study. Clinics. 2016;71(3):144–151

4. What was the indication of gastric lavage in this study?

Gastric lavage is indicated for immediate removal of stomach contents after an orally ingested overdose or poisoning in the absence of contraindications. It should be considered only where there is a risk of significant toxicity or imminent fatality and in situations of antidotal or other supportive modalities are inadequate (1). The use of this method in about 17.0% of the cases, in this study, was most probably from this perspective. On the other hand, it is recommended not to routinely employ this procedure in the management of poisoned patients because its efficiency of removal of the poisoning agents was found highly variable and diminished with time in experimental studies and clinical studies showed limited beneficial effects. Moreover, the procedure is associated with serious risks including hypoxia, dysrhythmias, perforation of the GI tract or pharynx, aspiration pneumonitis, and others (2). Therefore, consideration should first be given to other less invasive modalities of gastrointestinal decontamination such as activated charcoal or whole bowel irrigation (1).

1. Jenny J. Lu. Gastric Lavage. Emergency Medicine Procedures., edited by Eric F. Reichman, McGraw-Hill Education, 2013, 393-398

2. Vale JA, Kulig K; American Academy of Clinical Toxicology; European Association of Poisons Centres and Clinical Toxicologists. Position paper: gastric lavage. J Toxicol Clin Toxicol. 2004;42(7):933-943. 

5. What was the indication of frusemide and vitamin B complex?

Less frequently, diuretics can be used in forced diuresis, a method that uses volume loading to reduce tubular reabsorption, to actively eliminate harmful chemicals. For this aim, clinicians use loop diuretics like frusemide (1). Therefore, the use of frusemide in this study is consistent with current clinical practice.

1. StatPearls [Internet]. Therapeutic Uses of Diuretic Agents. Accessed on November 8, 2022. https://www.ncbi.nlm.nih.gov/books/NBK557838/

Vitamin B complex preparations were used in about 3.6% of the cases. Previous studies demonstrated favorable effects of vitamin B complex therapy on organophosphate-compound induced delayed neuropathy and alcoholic polyneuropathy (1, 2). The use of vitamin B complex preparations in our cases may be stemmed from these perspectives. 

1. Piao F, Ma N, Yamamoto H, Yamauchi T, Yokoyama K. Effects of prednisolone and complex of vitamin B1, B2, B6 and B12 on organophosphorus compound-induced delayed neurotoxicity. J Occup Health. 2004;46(5):359-364. 

2. Peters TJ, Kotowicz J, Nyka W, Kozubski W, Kuznetsov V, Vanderbist F et al. Treatment of alcoholic polyneuropathy with vitamin B complex: a randomised controlled trial. Alcohol Alcohol. 2006;41(6):636-642.

6. Were there admission cases to the hospital without IV fluid in this study? 

Yes, Patients who were free of signs and symptoms of toxicity necessitating fluid resuscitation were not give IV fluid. Such patients were admitted and given other treatments (symptomatic or antidotal) and followed, without giving IV fluid. 

7. What kind of laboratory tests in poisoning test do you have in this study? CBC, ECG, or blood toxicant concentration? What was the indication of laboratory test in poisoning cases?

The kind of tests that were used in the diagnosis process include CBC, ECG, Liver function tests (eg, alanine aminotransferase (ALT), aspartate aminotransferase AST), but not blood toxicant concentration. There were no methods used to determine the toxicant concentration in the set ups of the hospitals included this study. The poisoning managements were mainly guided by the clinical symp

---

## [Decision Letter · Decision Letter 1]

1 Mar 2023

PONE-D-22-26894R1Poisoning Cases and Their Management in Amhara Region, Ethiopia: Hospital Based Prospective StudyPLOS ONE

Dear Dr. Asrie,

Thank you for submitting your manuscript to PLOS ONE. After careful consideration, we feel that it has merit but does not fully meet PLOS ONE’s publication criteria as it currently stands. Therefore, we invite you to submit a revised version of the manuscript that addresses the points raised during the review process.

Unfortunately the two reviewers from the first round were not available to reassess your manuscript, so we have sought input from an additional reviewer whose report can be found below. As you will see from the comments, there remain significant concerns relating to the framing of the study within the body of existing literature and the reporting of the methodology which must be addressed before your manuscript is suitable for publication.

We look forward to receiving your revised manuscript.

Kind regards,

Dr Joseph Donlan

Senior Editor

PLOS ONE

Reviewers' comments:

Reviewer's Responses to Questions

**Comments to the Author**

1. If the authors have adequately addressed your comments raised in a previous round of review and you feel that this manuscript is now acceptable for publication, you may indicate that here to bypass the “Comments to the Author” section, enter your conflict of interest statement in the “Confidential to Editor” section, and submit your "Accept" recommendation.

Reviewer #3: (No Response)

2. Is the manuscript technically sound, and do the data support the conclusions?

Reviewer #3: Partly

3. Has the statistical analysis been performed appropriately and rigorously? 

Reviewer #3: No

4. Have the authors made all data underlying the findings in their manuscript fully available?

Reviewer #3: Yes

5. Is the manuscript presented in an intelligible fashion and written in standard English?

Reviewer #3: No

6. Review Comments to the Author

Reviewer #3: The article still has residual errors in grammar, I would suggest the authors use Grammarly or tools within their word processor.

Poisoning in emerging countries is a significant problem. It seems to me that his article needs to be clear upon its focus. It is long and information is repeated especially in the results section

It is important to describe the nature poisons being seen which this article does well. What is lacking is clearer description of the clinical setting such as rates of inter hospital transfer, antidote supply, health staff training and resources for advanced supportive care. This provides a context for the treatment and the results being provided. Given the breadth of significant toxicity I found the discussion of treatment rationale to be a bit superficial and in places outside established guidelines for treatment in these settings. Your article may describe the treatment your patients received but it is not an article about how to treat poisoned patients

Specific comments

Abstract

Poisoning is a significant public health problem globally and cases are being increased from time to time. As the advances in technological and social developments have led to ease accessibility of most drugs and chemical substances in the society, the number of incidences of poisoning can be higher. Ethiopia is one of the low-income countries that share its considerable burden of poisoning incidences and deaths.

Suggest

Poisoning is a significant public health problem globally. Ethiopia is a low-income country undergoing technological and social change that may increase access to drugs and chemicals potentially increasing the incidence of poisoning. This study describes the epidemiology of hospital admissions due to poisoning in a region of Ethiopia

Methods: Include a time period eg 2 years

Results

Some reorganisation of the order of the text would be useful and shortening as much of this is repeating the main results section

Eg

There were ZZZ cases with XX deaths. Almost all poisoning cases were intentional self- poisonings The most frequent poisonings were Organophosphate compounds, 145 (32.8%), and metal phosphides (majorly aluminum phosphide), 115 (26.0%). The ingested poison was most frequently accessed from the patients home,243 (55.0%), followed by purchase from local shops, 159 (36%)

The median duration of admission was XX. YY% required ventilation, the most poisoned case requiring ventilation was (insert compound) 43/78 deaths were due to aluminium phosphide

From the multivariate logistic regression analysis, age, altered level of consciousness on hospital arrival, distance from the area where the poisoning incidence happened to the hospital they attended, and no laboratory result as a part of the diagnosis process or investigation of the extent of toxicity were found to be significantly associated with the likelihood of poor treatment outcome or death.

Main Article

Background

I think this could be shortened, the first two paragraphs don’t add much to the background

The paragraph

“Poisoning is a common cause for emergency visit and hospital admission worldwide and a major reason for illness and death in many countries (3). The incidence of poisoning could be intentional or unintentional, and in children, because of the desire for imitating adults, the unintentional or accidental poisoning is frequent among them (4).”

Comment: most hospital admissions for poisoning are intentional and in the context of deliberate self harm. The second sentence regarding children is not supported by the reference nor is it correct that the motivation is a “desire for imitating adults”. In very young children accidental poisoning is common but in ages greater than 12 years it is commonly deliberate self harm with a wide range of precipitating factors including domestic violence, sexual assault, relationship difficulties etc I think this sentace is probably not required

Consider

“Intentional self-poisoning is a common cause for emergency visit and hospital admission worldwide and a major reason for illness and death in many countries (3).” And join this with the subsequent paragraph starting poisoning from pesticides

Methods

Please clarify are these 4 referral hospitals the only referral hospitals in the region? What population size do they serve? Are there specific transfer criteria from smaller hospitals for example are all poisoning cases transferred or only severe ones…..if it is the later then that referal bias needs to be discussed later

Results

Most of the result data sits in the tables, you should not repeat those results in the text rather you can provide a shorter overview of the table. In the discussion section you have an opportunity to bring the various themes together

Socio-demographic

This is well described in the table, your text is really just repeating the information in the table. So try rephrasing the text into a succinct overview of the table. Eg the majority of the patients were under 40 years of age, female, Christian of Amhara ethnicity see table 1

Distance and time of arrival to hospital

Please clarify is the primary hospital that they have presented to (ie not the referal hospital)

What percentage were direct admissions to the referral hospital versus transfer…this may have impact on outcomes as well as data interpretation

(see: Senarathna L, Buckley NA, Jayamanna SF, Kelly PJ, Dibley MJ, Dawson AH. Validity of referral hospitals for the toxicovigilance of acute poisoning in Sri Lanka. Bulletin of the World Health Organization. 2012;90:436-43a.)

Signs and symptoms

I am not sure what this section adds. The signs and symptoms will often depend upon the poisoning….the frequency of signs and symptoms with various poisons is well described in the literature. It may be more useful to characterise the patients using a poison severity score , or perhaps need for ventilation

Persson HE, Sjöberg GK, Haines JA, de Garbino JP. Poisoning severity score. Grading of acute poisoning. Journal of Toxicology: Clinical Toxicology. 1998 Jan 1;36(3):205-13.

Davies JO, Eddleston M, Buckley NA. Predicting outcome in acute organophosphorus poisoning with a poison severity score or the Glasgow coma scale. QJM: An International Journal of Medicine. 2008 May 1;101(5):371-9.

Types of poisoning

The text here is also just duplicating the table. I would suggest a summary statement.

In the table you could potentially add the deaths attributed to each poisoning ie the data that is in table 8

Dosage form of the poisons encountered and the routes of exposure

Reasons for poisoning incidents

Again the text is just duplicating the tables in these sections

Treatment

Probably the most important area of treatment is the need for advanced supportive care such as ventilation. Such treatment has important resource implications. Move the section from the discussion to the results

You mention gastric lavage but there is no mention of activated charcoal as part of the decontamination.

The subsequent list of treatment would be more informative if it focused upon recognised antidotal treatment eg atropine and perhaps calcium and magnesium. The rest of the drugs mentioned to not seem to specifically relate to poisoning

Duration of hospital stay

Time is a continuous variable, it is not clear to me why you have moved into a categorical variable, if you have the original data it may worth while in analysing it as continuous data especially for death

In the predictor of death analysis I think you should include the major poison groups, clinically it is likely that coma is driven by with organophophate or aluminium phosphide toxicity. Type of poisoning may be significant for the distance from primary hospital which would be a surrogate for delay to treatment. An adjusted analysis may help explain this

Discussion

In the discussion you need to discuss the potential referal bias as a potential limitation.

In treatment it would be useful to have some discussion about supply of medications, do you have shortages, how does your antidote stock compare with say WHO essential antidote list. What training do your medical staff have specifically for treatment of poisoning. I think it would be more useful to focus upon these bigger general questions than the discussion of individual drugs. Especially as many of the drugs you are suggesting to have a role in poisoning treatment are not mainstream and don’t feature in evidence based guidelines.

Within the discussion you mention activated charcoal but this doesn’t appear to have been used in your patients

The need for intubation and ventilation should move into the results

Referencing

Many of your references are old and have been superceded by bigger studies or systematic reviews….this is particularly the case with treatment

7. PLOS authors have the option to publish the peer review history of their article (what does this mean?). If published, this will include your full peer review and any attached files.

Reviewer #3: No

---

## [Author Response · Author response to Decision Letter 1]

5 May 2023

Response to Reviewer 

PONE-D-22-26894R1

Poisoning Cases and Their Management in Amhara Region, Ethiopia: Hospital Based Prospective Study PLOS ONE

Dear Dr. Asrie,

Thank you for submitting your manuscript to PLOS ONE. After careful consideration, we feel that it has merit but does not fully meet PLOS ONE’s publication criteria as it currently stands. Therefore, we invite you to submit a revised version of the manuscript that addresses the points raised during the review process.

Unfortunately the two reviewers from the first round were not available to reassess your manuscript, so we have sought input from an additional reviewer whose report can be found below. As you will see from the comments, there remain significant concerns relating to the framing of the study within the body of existing literature and the reporting of the methodology which must be addressed before your manuscript is suitable for publication.

All the above items have been included in this submission. 

If applicable, we recommend that you deposit your laboratory protocols in protocols.io to enhance the reproducibility of your results. Protocols.io assigns your protocol its own identifier (DOI) so that it can be cited independently in the future. For instructions see: https://journals.plos.org/plosone/s/submission-guidelines#loc-laboratory-protocols. Additionally, PLOS ONE offers an option for publishing peer-reviewed Lab Protocol articles, which describe protocols hosted on protocols.io. Read more information on sharing protocols athttps://plos.org/protocols?utm_medium=editorialemail&utm_source=authorletters&utm_campaign=protocols.

We look forward to receiving your revised manuscript.

Kind regards,

Dr Joseph Donlan

Senior Editor

PLOS ONE

Thank you dear editor for handling the peer review process of this manuscript. 

Reviewers' comments:

Reviewer's Responses to Questions

Comments to the Author

1. If the authors have adequately addressed your comments raised in a previous round of review and you feel that this manuscript is now acceptable for publication, you may indicate that here to bypass the “Comments to the Author” section, enter your conflict of interest statement in the “Confidential to Editor” section, and submit your "Accept" recommendation.

Reviewer #3: (No Response)

2. Is the manuscript technically sound, and do the data support the conclusions?

Reviewer #3: Partly

3. Has the statistical analysis been performed appropriately and rigorously?

Reviewer #3: No

We think that the statistical analysis has been improved during modifications based on a specifc comment in this regard (in revier comments to the author below). 

4. Have the authors made all data underlying the findings in their manuscript fully available?

Reviewer #3: Yes

5. Is the manuscript presented in an intelligible fashion and written in standard English?

Reviewer #3: No

We have made several modifications for language correcs through out the whole paper. 

Response to Reviwer #3

Dear reviewer, 

Thank you very much for your interesting and valuable comments, which are important to improve the quality of the manuscript to the highest standard possible. The comments have been addressed as follows. 

Reviewer #3: The article still has residual errors in grammar, I would suggest the authors use Grammarly or tools within their word processor.

We have made modifications in different parts of the paper for grammatical and/or typographical errors using “quillbot grammar checker” online software. 

Poisoning in emerging countries is a significant problem. It seems to me that his article needs to be clear upon its focus. It is long and information is repeated especially in the results section.

Thank you!

Taking this comment in account, we have reframed the contents of the manuscript in such a way that the focus to be on:

− type of poisoning agents encountered

− reasons for poisoning

− treatments given

− patient outcomes

− potential factors associated with poor outcome (death)

Considering the comment, we have removed some contents from the result section that we feel repeated or are not as such relevant. 

It is important to describe the nature poisons being seen which this article does well. What is lacking is clearer description of the clinical setting such as rates of inter hospital transfer, antidote supply, health staff training and resources for advanced supportive care. This provides a context for the treatment and the results being provided. Given the breadth of significant toxicity I found the discussion of treatment rationale to be a bit superficial and in places outside established guidelines for treatment in these settings. Your article may describe the treatment your patients received but it is not an article about how to treat poisoned patients

Right! We fully admitted what has been raised about what is lacking in this work. 

Regarding hospital transfer, the hospitals accept both referral and direct admission cases. However, this study did not segregate patients as direct admissions and transferred (referral) cases. We have already mentioned this in the discussion section as a limitation of the study. 

…antidote supply, health staff training and resources for advanced supportive care

This study is the first phase to be followed by poisoning treatment guideline preparation, establishing poisoning information centers, and preparing training manuals and delivering trainings to the health workers in the hospitals selected and others also. We planned to address the above mentioned issues just before these activities provided that we could secure a fund. Otherwise, even though we can’t support with data at this time and include in the paper, we know from our context that: 

− there is no strong antidote supply system

− on job trainings on poisoning and poisoning management are rare and there are few/no health professionals specialized with poisoning management or emergency medicine

− no separate emergency departments for poisoning cases

− there is scarcity of advanced supportive care facility in general and no specific advanced supportive care facilities dedicated for poisoning cases

So we may address these issues in a type of operational research or recommend to be addressed by other bodies including other scholars, the health facilities themselves or health bureau of the region. 

Specific comments

Abstract

Poisoning is a significant public health problem globally and cases are being increased from time to time. As the advances in technological and social developments have led to ease accessibility of most drugs and chemical substances in the society, the number of incidences of poisoning can be higher. Ethiopia is one of the low-income countries that share its considerable burden of poisoning incidences and deaths.

Suggest

Poisoning is a significant public health problem globally. Ethiopia is a low-income country undergoing technological and social change that may increase access to drugs and chemicals potentially increasing the incidence of poisoning. This study describes the epidemiology of hospital admissions due to poisoning in a region of Ethiopia. 

Thank you! This part has been modified as per this suggestion. 

Methods: Include a time period eg 2 years

The time period of the study was one year, and included there in the abstract. 

Results

Some reorganisation of the order of the text would be useful and shortening as much of this is repeating the main results section

Eg

There were ZZZ cases with XX deaths. Almost all poisoning cases were intentional self- poisonings The most frequent poisonings were Organophosphate compounds, 145 (32.8%), and metal phosphides (majorly aluminum phosphide), 115 (26.0%). The ingested poison was most frequently accessed from the patients home, 243 (55.0%), followed by purchase from local shops, 159 (36%)

The median duration of admission was XX. YY% required ventilation, the most poisoned case requiring ventilation was (insert compound) 43/78 deaths were due to aluminium phosphide

From the multivariate logistic regression analysis, age, altered level of consciousness on hospital arrival, distance from the area where the poisoning incidence happened to the hospital they attended, and no laboratory result as a part of the diagnosis process or investigation of the extent of toxicity were found to be significantly associated with the likelihood of poor treatment outcome or death.

Thank very much! We have adapted this text with some changes and incorporated in the manuscript (methods in the abstract). 

Main Article

Background

I think this could be shortened, the first two paragraphs don’t add much to the background

The paragraph

“Poisoning is a common cause for emergency visit and hospital admission worldwide and a major reason for illness and death in many countries (3). The incidence of poisoning could be intentional or unintentional, and in children, because of the desire for imitating adults, the unintentional or accidental poisoning is frequent among them (4).”

Comment: most hospital admissions for poisoning are intentional and in the context of deliberate self harm. The second sentence regarding children is not supported by the reference nor is it correct that the motivation is a “desire for imitating adults”. In very young children accidental poisoning is common but in ages greater than 12 years it is commonly deliberate self harm with a wide range of precipitating factors including domestic violence, sexual assault, relationship difficulties etc I think this sentace is probably not required

Consider

“Intentional self-poisoning is a common cause for emergency visit and hospital admission worldwide and a major reason for illness and death in many countries (3).” And join this with the subsequent paragraph starting poisoning from pesticides

Thank you! Based on the comments, the first paragraph has been deleted and the sentence about self-poisoning has been merged with the second paragraph (page 3). 

Methods

Please clarify are these 4 referral hospitals the only referral hospitals in the region? What population size do they serve?

Are there specific transfer criteria from smaller hospitals for example are all poisoning cases transferred or only severe ones…..if it is the later then that referral bias needs to be discussed later

There were 5 referral hospitals in the region during the study period and 1 specialized hospital was inaugurated lately. Accordingly, there are currently 6 referral hospitals in the region. Four of the referral hospitals were conveniently selected as the study site. 

The locations of the hospitals and the catchment area populations they serve have been included in the Methods section (The study area, page 5). 

With the current practice, the major criteria (reason) to transfer cases from smaller/primary hospitals to referral hospitals is the severity of the poisoning and/or unsatisfactory response to the treatments given. The referral hospitals also accept and treat cases that are not referred from the smaller hospitals. Accordingly, the poisoning cases in the referral hospitals include both referral cases and direct admissions. However, this study didn’t segregate the cases in this way and discussed as a limitation in the original submission. 

Results

Most of the result data sits in the tables, you should not repeat those results in the text rather you can provide a shorter overview of the table. In the discussion section you have an opportunity to bring the various themes together

Thank you for the suggestion. 

Socio-demographic

This is well described in the table, your text is really just repeating the information in the table. So try rephrasing the text into a succinct overview of the table. Eg the majority of the patients were under 40 years of age, female, Christian of Amhara ethnicity see table 1

Thank you! We have made changes in this part based on the comment. 

Distance and time of arrival to hospital

Please clarify is the primary hospital that they have presented to (ie not the referal hospital)

What percentage were direct admissions to the referral hospital versus transfer…this may have impact on outcomes as well as data interpretation

(see: Senarathna L, Buckley NA, Jayamanna SF, Kelly PJ, Dibley MJ, Dawson AH. Validity of referral hospitals for the toxicovigilance of acute poisoning in Sri Lanka. Bulletin of the World Health Organization. 2012;90:436-43a.)

Thank you for the concern and the reference provided!

The distances and times of arrival at the hospitals are regarding the referral hospitals considered in the study (not the primary hospitals). As we tried to indicate above, this study didn’t identify patients as direct admission or referral cases. 

Signs and symptoms

I am not sure what this section adds. The signs and symptoms will often depend upon the poisoning….the frequency of signs and symptoms with various poisons is well described in the literature. It may be more useful to characterise the patients using a poison severity score , or perhaps need for ventilation.

Persson HE, Sjöberg GK, Haines JA, de Garbino JP. Poisoning severity score. Grading of acute poisoning. Journal of Toxicology: Clinical Toxicology. 1998 Jan 1;36(3):205-13.

Davies JO, Eddleston M, Buckley NA. Predicting outcome in acute organophosphorus poisoning with a poison severity score or the Glasgow coma scale. QJM: An International Journal of Medicine. 2008 May 1;101(5):371-9.

Right! We removed the "Signs and Symptoms" section (Table 3 of the previous version) considering this comment. Instead, we kept the level of consciousness data. In each case it was determined using clinical judgment and especially in the patients with altered levels of consciousness using the Glasgow coma scale. Accordingly, the cases were classified as conscious, semiconscious, and unconscious. This result has been pres

---

## [Decision Letter · Decision Letter 2]

15 Nov 2023

PONE-D-22-26894R2Poisoning Cases and Their Management in Amhara Region, Ethiopia: Hospital Based Prospective StudyPLOS ONE

Dear Dr. Asrie,

Thank you for submitting your manuscript to PLOS ONE. After careful consideration, we feel that it has merit but does not fully meet PLOS ONE’s publication criteria as it currently stands. Therefore, we invite you to submit a revised version of the manuscript that addresses the points raised during the review process.

We look forward to receiving your revised manuscript.

Kind regards,

Jianhong Zhou

Staff Editor

PLOS ONE

**Additional Editor Comments:**

This revision is evaluated by two reviewers. Please see their comments below.

Reviewers' comments:

Reviewer's Responses to Questions

**Comments to the Author**

1. If the authors have adequately addressed your comments raised in a previous round of review and you feel that this manuscript is now acceptable for publication, you may indicate that here to bypass the “Comments to the Author” section, enter your conflict of interest statement in the “Confidential to Editor” section, and submit your "Accept" recommendation.

Reviewer #3: (No Response)

Reviewer #4: (No Response)

2. Is the manuscript technically sound, and do the data support the conclusions?

Reviewer #3: Partly

Reviewer #4: No

3. Has the statistical analysis been performed appropriately and rigorously? 

Reviewer #3: I Don't Know

Reviewer #4: No

4. Have the authors made all data underlying the findings in their manuscript fully available?

Reviewer #3: No

Reviewer #4: Yes

5. Is the manuscript presented in an intelligible fashion and written in standard English?

Reviewer #3: No

Reviewer #4: No

6. Review Comments to the Author

Reviewer #3: You need be more specific about design, although it is not explicitly stated it appears that you targeted a total of 440 patients recruiting 110 in each site. If that is the case you should say that explicitly including whether all consecutive patients were approached, how many refused (and include some basic information about the refusal such as the poison they were exposed to , gender and age) Further how Lon did it take you in each site to recruit the patients eg how many weeks

Reviewer #4: Manuscript ID: PONE-D-22-26894R2

Thank you for the opportunity to review this manuscript entitled: “Poisoning Cases and Their Management in Amhara Region, Ethiopia: Hospital Based Prospective Study”

There are still numerous errors relating to grammar, syntax and academic one. I suggest that the authors employ the services of a professional English language editor with a background in medical journal articles

In the Methods section of the Abstract it is stated that data collection was from Jan to Dec 2028, which implies that a convenience sampling was used. However, in the main text under Methods, the authors indicate that sample size determination was used. This is conflicting and needs to be clarified.

Overall, there are too many tables and figures. Most of the data can be included in 2-3 tables. Also, there is a lot of repetition between the text and figures/tables. Table 3: Seems like “Died” and “Cured” under “Treatment outcome” have been incorrectly labelled. The authors are encouraged to read through other similar publications which will assist them with improving the layout and design of this article. The discussion section is too long.

7. PLOS authors have the option to publish the peer review history of their article (what does this mean?). If published, this will include your full peer review and any attached files.

Reviewer #3: No

Reviewer #4: No

---

## [Author Response · Author response to Decision Letter 2]

7 Dec 2023

Response to Reviewers

Manuscript Title: Poisoning Cases and Their Management in Amhara Region, Ethiopia: Hospital-Based Prospective Study

Manuscript Submission Number: PONE-D-22-26894R2

Journal: PLOS ONE

Dear Editor, 

We are very thankful to you for handling the publication process of this manuscript. 

Modified portions or newly added texts in the revision process are marked-up with greed color, whereas deletions are shown by red colored texts with Strikethrough line. 

Response to Reviewer #3

Dear Reviewer, 

We are grateful for your dedication and time spent in reviewing this manuscript. The comments forwarded are greatly appreciated and used to help improve the manuscript. They have been addressed as follows.

Reviewers' comments:

Reviewer's Responses to Questions

Comments to the Author

1. If the authors have adequately addressed your comments raised in a previous round of review and you feel that this manuscript is now acceptable for publication, you may indicate that here to bypass the “Comments to the Author” section, enter your conflict of interest statement in the “Confidential to Editor” section, and submit your "Accept" recommendation.

Reviewer #3: (No Response)

2. Is the manuscript technically sound, and do the data support the conclusions?

Reviewer #3: Partly 

Thank you!

By taking into account the feedback you sent we have made changes to improve the manuscript in this regard. We include additional points in the conclusion part. 

3. Has the statistical analysis been performed appropriately and rigorously?

Reviewer #3: I Don't Know

As the result is dichotomous, i.e., survived or died, we have performed binary logistic regression adjusted to potential factors that might affect patient outcome (as cured or died). First we performed bivariate logistic regression to identify potential predictors of the patient outcome. Then the factors found to significantly associate with probability of death in bivariate analysis were included in multivariate analysis. We performed backward stepwise elimination logistic regression and we selected the best model among the alternative models found in the analysis. 

4. Have the authors made all data underlying the findings in their manuscript fully available?

Reviewer #3: No

We think that we made sufficient data underlying the findings available in the manuscript. Moreover, the full data set was submitted to the journal during manuscript submission. However, as another reviewer commented that there are too many tables and figures and the result could be summarized in 2-3 tables, 2 tables and 2 figures have been removed in this version. 

5. Is the manuscript presented in an intelligible fashion and written in standard English?

Reviewer #3: No

Thanks! We have reviewed the entire document and made a number of changes to fix grammatical errors. We corrected some grammatical errors using QuillBot online grammar checker. Additionally, the document was revised for language clarity by Dr. Yoseph, a senior English language professional (PhD in Teaching English as a Foreign Language (TEFL)) at University of Gondar, and further corrections were made based on his recommendation. 

6. Review Comments to the Author

Reviewer #3: You need be more specific about design, although it is not explicitly stated it appears that you targeted a total of 440 patients recruiting 110 in each site. If that is the case you should say that explicitly including whether all consecutive patients were approached, how many refused (and include some basic information about the refusal such as the poison they were exposed to , gender and age) Further how Lon did it take you in each site to recruit the patients eg how many weeks

Thank you for your concern! 

The number of cases targeted was determined to be 442 (based on sample size calculation, page 4 of the previous version of the manuscript). And then, the subjects were divided to the four hospitals considered in this study. 

− University of Gondar Comprehensive Specialized Hospital = 110 patients

− Felege Hiwot Referral Hospital = 110 patients

− Debre Markos Referral Hospital = 111 patients 

− Dessie Referral Hospital = 111 patients

The number of cases (110 or 111) to be targeted from each site was determined simply by lottery method. 

From preliminary assessment we made before the data collection, we had been informed that irregular numbers of poisoning cases attend each hospital. Referring log books in the emergency departments and communicating the healthcare professionals working in the emergency department of each hospital, we had been informed that the number of poisoning cases attending the departments in each year is unpredictable and roughly ranges 80 to 130 patients per year. Because of this we couldn’t approach randomly selected cases and rather we decided to and approached all consecutive cases until we got the number of cases taken from each hospital. Very few cases refused to participate in the study (refusal was so rare) and had been excluded from the study. In addition, very few cases from whom adequate data were not obtained had been excluded from the study. However, we collected no data from such cases so that we have no information about them at hand. 

The data collection was started on the same date at all sites, whereas there were some differences in the length of time needed to finish the data collection at each site. In all hospitals, the data collection was started on January 01, 2018 and completed in December 2018 on different specific dates shown below. 

− University of Gondar Comprehensive Specialized Hospital = on December 26, 2018

− Felege Hiwot Referral Hospital = on December 24, 2018 

− Debre Markos Referral Hospital = on December 21, 2018

− Dessie Referral Hospital = on December 29, 2018

It was taken almost a year to complete the data collection at each site. For this reason, we stated that the data collection period ran from January to December of 2018 (to collect the data from the total of 442 cases). 

Response to Reviewer #4

Dear Reviewer, 

Thank you for your time and effort invested in reviewing this manuscript. The comments raised are highly valued and used as input to improve the manuscript. They have been addressed as follows:

Reviewers' comments:

Reviewer's Responses to Questions

Comments to the Author

1. If the authors have adequately addressed your comments raised in a previous round of review and you feel that this manuscript is now acceptable for publication, you may indicate that here to bypass the “Comments to the Author” section, enter your conflict of interest statement in the “Confidential to Editor” section, and submit your "Accept" recommendation.

Reviewer #4: (No Response)

2. Is the manuscript technically sound, and do the data support the conclusions?

Reviewer #4: No

Thank you!

We think that this issue has been improved through addressing the feedbacks you provided under the "Review comments to Author". 

3. Has the statistical analysis been performed appropriately and rigorously?

Reviewer #4: No

As the outcome measure is dichotomous, i.e., survived or died, we have performed binary logistic regression adjusted to potential factors that might affect the outcome (as cured or died). Initially, we performed bivariate logistic regression to identify potential predictors of the patient outcome. Then the factors that were found to be significantly associated with the probability of death in bivariate analysis were included in multivariate analysis. We performed backward stepwise elimination logistic regression and we selected the best model among the alternative models found in the analysis.

4. Have the authors made all data underlying the findings in their manuscript fully available?

Reviewer #4: Yes

5. Is the manuscript presented in an intelligible fashion and written in standard English?

Reviewer #4: No

We reviewed the manuscript to increase the clarity and ease of understandability. Accordingly, we made many changes at various sections in the process of revision. 

6. Review Comments to the Author

Reviewer #4: Manuscript ID: PONE-D-22-26894R2

Thank you for the opportunity to review this manuscript entitled: “Poisoning Cases and Their Management in Amhara Region, Ethiopia: Hospital Based Prospective Study”

There are still numerous errors relating to grammar, syntax and academic one. I suggest that the authors employ the services of a professional English language editor with a background in medical journal articles

We have gone through the whole document and made numerous amendments in this regard. First we made corrections on grammatical errors using “quillbot online grammar checker”. Additionally, the document was revised for language clarity by Dr. Yoseph, a senior English language professional (PhD in Teaching English as a Foreign Language (TEFL)) at University of Gondar, and further corrections were made based on his recommendation. The manuscript has been also revised for academic issues by Dr. Zewdneh Shewamene Sabe (working at Population Council, Ethiopia) and additional corrections were made (eg, deletion of portions in the result and discussion parts). But we are unable to get professional language editor with a background in medical journal articles. 

In the Methods section of the Abstract it is stated that data collection was from Jan to Dec 2028, which implies that a convenience sampling was used. However, in the main text under Methods, the authors indicate that sample size determination was used. This is conflicting and needs to be clarified.

Thank you!

First, we determined the number of cases to be included in the study by calculating the sample size as stated in the manuscript (not conveniently determined). Then, we distributed the 442 samples to the four study sites; 111 cases to each of two hospitals and 110 cases to each of the other two hospitals. After this, we started data collection and the period stated is the time required to collect the data in each hospital. 

At all sites, the data collection was started on January 1, 2018 and completed in the fourth week of December 2018 on different specific dates. For this reason, we stated that the data collection period ran from January to December of 2018. The time was not predetermined before the data collection. We determined the sample cases to be considered first, distributed them to the four study sites. It took the stated period (January to December 2018) to complete the data collection at each site. 

As of the information we got from the emergency departments of the hospitals before starting the data collection, the cases flow was not predictable. Because of this we couldn’t follow a specific sampling technique and rather we decided to consider all consecutive cases till we got the specified number of patients in each hospital. There were few refusal cases and few others with incomplete data and excluded from the study. However, since the data was collected until we achieved the predetermined number of subjects in each study site; the total cases targeted are equal to the originally calculated sample size. 

Overall, there are too many tables and figures. Most of the data can be included in 2-3 tables. Also, there is a lot of repetition between the text and figures/tables. 

Thank you!

In consideration of this comment, 2 tables and 2 figures that we feel are redundant or do not contain as much relevant information have been removed. The discussion points pertaining to the contents of these tables and figures have also been eliminated. These are the figures and tables that have been removed. 

− Table 2 on Page 7…Variable included in the multivariate analysis 

− Table 4 on page 8…We think the content is not as much relevant.

− Figure 2 on page 11…The variable included in the multivariate analysis

− Figure 3 on page 12….Because the content appeared in the multivariate analysis result table. 

Table 3: Seems like “Died” and “Cured” under “Treatment outcome” have been incorrectly labelled. The authors are encouraged to read through other similar publications which will assist them with improving the layout and design of this article. The discussion section is too long. 

Thank you!

We replaced the term “died” by “cured” and the vice versa. 

The discussion part has been reduced from 5 and half pages to about 4 pages. 

Please note that:

Texts with red font color and strikethrough line – indicate deletion in the process of revision. 

Texts with green highlight – indicate newly added texts in the revision process.

---

## [Decision Letter · Decision Letter 3]

29 Feb 2024

PONE-D-22-26894R3Poisoning Cases and Their Management in Amhara Region, Ethiopia: Hospital-Based Prospective StudyPLOS ONE

Dear Dr. Asrie,

Thank you for submitting your manuscript to PLOS ONE. After careful consideration, we feel that it has merit but does not fully meet PLOS ONE’s publication criteria as it currently stands. Therefore, we invite you to submit a revised version of the manuscript that addresses the points raised during the review process.

Comments from PLOS Editorial Office: We note that one or more reviewers has recommended that you cite specific previously published works. As always, we recommend that you please review and evaluate the requested works to determine whether they are relevant and should be cited. It is not a requirement to cite these works. We appreciate your attention to this request.

We look forward to receiving your revised manuscript.

Kind regards,

Suzit Bhusal

Academic Editor

PLOS ONE

Journal Requirements:

Reviewers' comments:

Reviewer's Responses to Questions

**Comments to the Author**

1. If the authors have adequately addressed your comments raised in a previous round of review and you feel that this manuscript is now acceptable for publication, you may indicate that here to bypass the “Comments to the Author” section, enter your conflict of interest statement in the “Confidential to Editor” section, and submit your "Accept" recommendation.

Reviewer #3: (No Response)

Reviewer #5: All comments have been addressed

2. Is the manuscript technically sound, and do the data support the conclusions?

Reviewer #3: Yes

Reviewer #5: Yes

3. Has the statistical analysis been performed appropriately and rigorously? 

Reviewer #3: Yes

Reviewer #5: Yes

4. Have the authors made all data underlying the findings in their manuscript fully available?

Reviewer #3: Yes

Reviewer #5: Yes

5. Is the manuscript presented in an intelligible fashion and written in standard English?

Reviewer #3: No

Reviewer #5: (No Response)

6. Review Comments to the Author

Reviewer #3: There are still some errors in English or confusing expressions, in some places repetition. I have marked up your tracked document in yellow and also suggested strike thru to address this.

In the results you need to detail how many patients were approached and how many refused consent and were subsequently excluded.

Your paper and discussion is really about the epridemioogy of poisoning and some service utilisation. In this context I would avoid to much discussion around the rationale for treatments inevitably it will be a bit superficial.

In the discussion the use of exisiting medicines for antidotes is rarely approved it is more commonly recommended in guidelines based on the level of evidence. As this is an epidimology paper you probably do not need to discuss the evidence for the role of magnesium.

For magnesium I have suggested shortening that section. Alos a more approbate reference if needed is

Brvar M, Chan MY, Dawson AH, Ribchester RR, Eddleston M. Magnesium sulfate and calcium channel blocking drugs as antidotes for acute organophosphorus insecticide poisoning–a systematic review and meta-analysis. Clinical Toxicology. 2018 Aug 3;56(8):725-36.

Reviewer #5: All the comments have been addressed. The manuscript can be accepted for publication. No further changes required

7. PLOS authors have the option to publish the peer review history of their article (what does this mean?). If published, this will include your full peer review and any attached files.

Reviewer #3: No

Reviewer #5: No

---

## [Author Response · Author response to Decision Letter 3]

22 Mar 2024

Response to Reviewer #3 

Dear Reviewer, 

We want to express our sincere gratitude for your thorough review of our paper. Your insights and feedback were incredibly valuable to us. Your constructive feedback helped us to see points for improvement and act accordingly. We truly appreciate once again for the time and effort you put into providing such thoughtful feedback. The points raised have been addressed as follows.

Please note that newly added or modified texts during the revision process are indicated with a green highlight, while texts that have been deleted are indicated with a red font color and a strikethrough line.

Reviewers' comments:

Reviewer's Responses to Questions

Comments to the Author

1. If the authors have adequately addressed your comments raised in a previous round of review and you feel that this manuscript is now acceptable for publication, you may indicate that here to bypass the “Comments to the Author” section, enter your conflict of interest statement in the “Confidential to Editor” section, and submit your "Accept" recommendation.

Reviewer #3: (No Response)

2. Is the manuscript technically sound, and do the data support the conclusions?

Reviewer #3: Yes 

Thank you!

3. Has the statistical analysis been performed appropriately and rigorously?

Reviewer #3: Yes

Thank you!

4. Have the authors made all data underlying the findings in their manuscript fully available?

Reviewer #3: Yes

Thank you!

5. Is the manuscript presented in an intelligible fashion and written in standard English?

Reviewer #3: No

Thank you! We have gone over the entire manuscript document again and made some revisions to address some grammatical errors and clarity issues, with emphasis on the suggestions you provided in the 'Review Comments to the Author' section below.

6. Review Comments to the Author

Reviewer #3: There are still some errors in English or confusing expressions, in some places repetition. I have marked up your tracked document in yellow and also suggested strike thru to address this.

Thank you for marking up specific sentences and texts that need more attention for language correction. We have gone through the manuscript and made some modifications based on this suggestion.

In the results you need to detail how many patients were approached and how many refused consent and were subsequently excluded.

Thank you! 

A small number of cases declined consent to participate in the study and were consequently excluded. Additionally, some cases from whom sufficient data could not be obtained were excluded from the study. Thus, we have included a description in the first part of the results section, stating how many patients were approached and how many refused consent and were subsequently excluded. We've also provided information on the number of cases that were excluded due to the incomplete data they provided. 

Your paper and discussion is really about the epridemioogy of poisoning and some service utilisation. In this context I would avoid to much discussion around the rationale for treatments inevitably it will be a bit superficial.

Thank you!

We acknowledged that it would be appropriate to maintain our primary emphasis on epidemiology and service utilization in discussing the results. We streamlined our discussion, ensuring that the discussion regarding the rationale behind treatments has been limited. However, there are still some points seemingly made about the rationale behind the use of gastric lavage as a GI decontamination method. We were advised by a reviewer during the first round of revision to include these points regarding gastric lavage.

In the discussion the use of exisiting medicines for antidotes is rarely approved it is more commonly recommended in guidelines based on the level of evidence. As this is an epidimology paper you probably do not need to discuss the evidence for the role of magnesium.

For magnesium I have suggested shortening that section. Alos a more approbate reference if needed is

Brvar M, Chan MY, Dawson AH, Ribchester RR, Eddleston M. Magnesium sulfate and calcium channel blocking drugs as antidotes for acute organophosphorus insecticide poisoning–a systematic review and meta-analysis. Clinical Toxicology. 2018 Aug 3;56(8):725-36.

Thank you!

We deleted the section as suggested. Besides, we found the recommended article appropriate and believed it would help consolidate our discussion. Thus, we have cited the paper in the manuscript as per the suggestion (ref. 23). 

Reviewer #5 

We would like to extend our heartfelt appreciation for taking the time to review our work. Your comments have been greatly valuable in improving the manuscript to the highest possible quality. Thank you once again for your highly considerable contribution to the review process of our work.

---

## [Decision Letter · Decision Letter 4]

25 Apr 2024

Poisoning cases and their management in Amhara National Regional State, Ethiopia: hospital-based prospective study

PONE-D-22-26894R4

Dear Dr. Asrie,

We’re pleased to inform you that your manuscript has been judged scientifically suitable for publication and will be formally accepted for publication once it meets all outstanding technical requirements.

Kind regards,

Senthil Kumaran, MBBS, MD, DNB

Academic Editor

PLOS ONE

Additional Editor Comments (optional):

Reviewers' comments:

Reviewer's Responses to Questions

**Comments to the Author**

1. If the authors have adequately addressed your comments raised in a previous round of review and you feel that this manuscript is now acceptable for publication, you may indicate that here to bypass the “Comments to the Author” section, enter your conflict of interest statement in the “Confidential to Editor” section, and submit your "Accept" recommendation.

Reviewer #3: All comments have been addressed

2. Is the manuscript technically sound, and do the data support the conclusions?

Reviewer #3: Yes

3. Has the statistical analysis been performed appropriately and rigorously? 

Reviewer #3: Yes

4. Have the authors made all data underlying the findings in their manuscript fully available?

Reviewer #3: Yes

5. Is the manuscript presented in an intelligible fashion and written in standard English?

Reviewer #3: Yes

6. Review Comments to the Author

Reviewer #3: (No Response)

7. PLOS authors have the option to publish the peer review history of their article (what does this mean?). If published, this will include your full peer review and any attached files.

Reviewer #3: No

---

## [Editor Report · Acceptance letter]

20 May 2024

PONE-D-22-26894R4 

PLOS ONE

Dear Dr. Asrie, 

I'm pleased to inform you that your manuscript has been deemed suitable for publication in PLOS ONE. Congratulations! Your manuscript is now being handed over to our production team.

Kind regards, 

on behalf of

Dr. Senthil Kumaran 

Academic Editor

PLOS ONE